# Residual Factorized Fourier Neural Operator for simulation of three-dimensional turbulence

## Abstract

Neural Operators, particularly Fourier Neural Operators (FNO), have proven highly effective in simulating partial differential equations (PDEs), such as the Navier-Stokes equations. We propose the Residual Factorized Fourier Neural Operator (Res-F-FNO) for simulating three-dimensional (3D) flows, specifically focusing on flow dynamics around a cube. We extend the Factorized Fourier Neural Operator (F-FNO) architecture by incorporating additional residual connections. This change effectively reintroduces small-scale dynamic flows that may be lost due to truncated Fourier modes, resulting in improved accuracy when modeling wind fields. Our proposed Res-F-FNO model surpasses the performance of the standard F-FNO, achieving an error reduction of over 30% in simulating 3D flows. Furthermore, we propose the concept of a skip-corrector, to address the problem of accumulated errors over multiple time steps. The skip-corrector was specifically trained to predict the behaviour of turbulences at a considerably extended time interval. Incorporating the skip-corrector into the prediction process reduces the average error in simulating 100 time steps by more than 50%. Additionally, we adopt a modified training approach in which random time steps are chosen as the initial condition for each sample in every epoch, as opposed to generating a dataset by propagating each sample across all time steps. This leads to a significant reduction in the the number of training iterations required for the models to achieve convergence.

## 1 Introduction

Complex partial differential equations (PDEs) play a significant role in numerous fields in science and engineering. These equations are instrumental in various applications, ranging from climate change modeling (Taylor et al., 2011) to seismic wave propagation analysis (Chapman, 2004), stress distribution exploration within materials (Pascon, 2019), and the investigation of biological processes like disease spread (Majid et al., 2021). To accurately capture the inherent complexities of real-world phenomena, numerical solvers necessitate a fine-grained discretization. This in turn, imposes considerable computational burdens and significant memory requirements (Hosseini et al., 2016). Among these, the Navier-Strokes equations stand out for their role in defining fluid flow characterstics and turbulence behavior. These equations can be solved numerically through computational fluid dynamics (CFD). Over the past few years, an increasing number of approaches based on neural networks have been developed to complement, accelerate, or even completely replace traditional CFD approaches for modeling turbulent flows (Brunton et al., 2020; Duraisamy et al., 2019; Um et al., 2021; Sirignano & Spiliopoulos, 2018).

Traditional neural network developments have been mainly focused on learning mappings between finite dimensional Euclidean spaces or finite sets. These models are good at learning a particular instance of the governing equation, but have difficulty generalizing when the function parameters, initial condition, or boundary conditions change (Kovachki et al., 2023). The Fourier Neural Operator (FNO) (Li et al., 2021) is of particular interest as it distinguishes itself from classical neural networks by learning mappings between infinite dimensional function spaces instead of finite dimensional Euclidean spaces. This approach enables the learning of families of PDEs with enhanced generalizability across various initial and boundary conditions. The Factorized Fourier Neural Op-

erator (F-FNO) (Tran et al., 2023) represents an advancement over the FNO, allowing architectures with significantly more layers to converge. As a result, it outperforms the FNO on a wide range of challenges, including the Navier-Stokes problem and the airfoil flow problem.

The existing research primarily focuses on forecasting two-dimensional (2D) turbulent flows (Wu et al., 2022; Cheng & Zhang, 2021; Peng et al., 2022a; Hasegawa et al., 2020a; Li et al., 2022b; Hasegawa et al., 2020b; Jin et al., 2021). Addressing the prediction of three-dimensional (3D) flows using deep neural networks poses significant challenges due to the substantial increase in simulation data volume, demanding greater computational resources and graphics memory for training. Accurately capturing the behavior of nonlinear 3D flows requires a considerably more complex neural network architecture, characterized by a substantially higher parameter count compared to the 2D scenario (Peng et al., 2022b). Particular approaches (Li et al., 2022a; Peng et al., 2022b; 2023), utilize FNOs to predict 3D turbulent flows and to simulate 3D dynamics of urban microclimate. Nonetheless, the investigation of small-scale 3D turbulent flows around objects remains unexplored.

In this paper, we propose the Residual Factorized Fourier Neural Operator (Res-F-FNO), which incorporates additional residual connections to enhance the accuracy of predicting 3D turbulent flow around a cube, surpassing the performance of the default F-FNO. Furthermore, we introduce the notion of a skip-corrector, embodied by a Res-F-FNO model, which effectively reduces the accumulated error over time, leading to enhanced accuracy in predicting a significantly larger number of time steps. Moreover, we detail a training strategy involving a limited number of samples with extended time intervals and a random starting condition within each sample.

Overall, we make the following three key contributions:

1. We present the Res-F-FNO, which significantly enhances the precision in forecasting 3D flows around a cube in comparison to F-FNO, achieved through the incorporation of additional residual connections (Eq. (7), Fig. 2, Fig. 3).

2. We propose a novel concept termed as skip-corrector, which effectively reduces accumulated errors over consecutive time step predictions (Eq. (8), Fig. 5).

3. We showcase an innovative training methodology utilizing fewer samples but varying initial conditions within each sample to train the Res-F-FNO model.

## 2 BACKGROUND AND RELATED WORK

The primary focus of recent research centers on the development of neural network based models, which directly learn the mapping between infinite-dimensional function spaces.Because these models do not rely on the grid resolution used during training, they can solve PDEs, like the Navier-Strokes equation, for different discretizations (Anandkumar et al., 2019; Kovachki et al., 2023; Li et al., 2021; 2022b; Tran et al., 2023; Li et al., 2020).

The goal is to learn a mapping between two infinite-dimensional function spaces based on a finite collection of input-output pairs obtained from this mapping. Let $D \subset \mathbb{R}^d$ be a bounded, open set and define the target (typically) non-linear map as $G^\dagger : \mathcal{A} \to \mathcal{U}$, where $\mathcal{A} = \mathcal{A}(D; \mathbb{R}^{d_a})$ and $\mathcal{U} = \mathcal{U}(D; \mathbb{R}^{d_u})$ be separable Banach spaces of functions taking values in $\mathbb{R}^{d_a}$ and $\mathbb{R}^{d_u}$ from that set $D$ respectively. Furthermore, suppose we have the input-output pairs $\{a_j, u_j\}_{j=1}^N$ where $a_j \sim \mu$ describes an i.i.d. sequence from the probability measure $\mu$ supported on $\mathcal{A}$ and $u_j = G^\dagger(a_j)$ is the output from the mapping possibly corrupted with noise. The aim is to build a neural network, which learns an approximation of $G^\dagger$ by constructing a parametric map

$$G : \mathcal{A} \times \Theta \to \mathcal{U} \qquad \text{or equivalently,} \qquad G_\theta : \mathcal{A} \to \mathcal{U}, \quad \theta \in \Theta \qquad (1)$$

for some finite-dimensional parameter space $\Theta$ by choosing $\theta^\dagger \in \Theta$ such that $G(\cdot, \theta^\dagger) = G_{\theta^\dagger} \approx G^\dagger$. In general $a_j$ and $u_j$ are functions. Therefore, to use them as input for a neural network model, they need to be evaluated point-wise. Let $D_n = \{x_1, \ldots, x_n\} \subset D$ be a $n$-point discretization of the domain $D$ and assume a finite collection of observed input-output pairs $a_j|_{D_n} \in \mathbb{R}^{n \times d_a}$, $u_j|_{D_n} \in \mathbb{R}^{n \times d_u}$, indexed by $j$. To be discretization-invariant and therefore a true function space method, the neural network must be capable of producing an answer $u(x)$ for any $x \in D$ given an input $a \sim \mu$. This property allows the transfer of solutions between different gird geometries and

discretizations (Li et al., 2022a; Peng et al., 2022b; Li et al., 2021; Tran et al., 2023; Peng et al., 2022a; Anandkumar et al., 2019).

**Neural Operator.** Anandkumar et al. (2019) proposed the concept of Neural Operators designed to approximate the mapping between function spaces and formulated as an iterative architecture $v_0 \mapsto v_1 \mapsto \cdots \mapsto v_T$ where $v_j$ for $j = 0, 1, \ldots, T - 1$ is a sequence of functions each taking values in $\mathbb{R}^{d_v}$. In a first step the input $a \in \mathcal{A}$ is lifted to a higher dimension $\mathbb{R}^{d_{v_0}}$ by a neural network layer. Subsequently, this higher dimensional representation is updated iteratively by

$$v_{t+1}(x) = \sigma(W v_t(x) + \mathcal{K}(a; \phi) v_t(x)), \qquad \forall x \in D \tag{2}$$

where $\mathcal{K} : \mathcal{A} \times \Theta_K \to \mathcal{L}(\mathcal{U}(D; \mathbb{R}^{d_v}), \mathcal{U}(D; \mathbb{R}^{d_v}))$ maps to bounded linear operators on $\mathcal{U}(D; \mathbb{R}^{d_v})$ and is parameterized by $\phi \in \Theta_{\mathcal{K}}$, $W : \mathbb{R}^{d_v} \to \mathbb{R}^{d_v}$ describes a linear transformation, and $\sigma : \mathbb{R} \to \mathbb{R}$ is an element-wise non-linear activation function. Anandkumar et al. (2019) define $\mathcal{K}(a; \phi)$ to be a kernel integral transformation parameterized by a neural network. Lastly, a point-wise function $\mathbb{R}^{d_{v_T}} \to \mathbb{R}^{d_u}$ maps the hidden representation $v_T$ to the output function $u$.

**Fourier Neural Operator (FNO).** Li et al. (2021) presented the Fourier Neural Operator, which replaces the kernel integral operator $\mathcal{K}(a; \phi)$ in Eq. (2) by a convolution operator defined in Fourier space. Rather than directly parameterizing the kernel in the Domain $D$, this approach contemplates its representation in Fourier space and parameterizes it there. Let $\mathcal{F}$ and $\mathcal{F}^{-1}$ denote the Fourier transform and its inverse transform of a function $f : D \to \mathbb{R}^{d_v}$ respectively. The kernel can then be defined as Fourier integral operator

$$(\mathcal{K}(\phi) v_t)(x) = \mathcal{F}^{-1}(R_\phi \cdot (\mathcal{F} v_t))(x), \qquad \forall x \in D \tag{3}$$

where $R_\phi$ is the Fourier transform of a periodic function $\kappa : \bar{D} \to \mathbb{R}^{d_v \times d_v}$ parameterized by $\phi \in \Theta_{\mathcal{K}}$. Assuming periodicity for $\kappa$ enables a Fourier series expansion, allowing the discretization of the frequency modes $k \in \mathbb{Z}^d$. By truncating the Fourier series at a maximum mode $k_{\max} = |Z_{k_{\max}}| = |\{k \in \mathbb{Z}^d : |k_j| \le k_{\max,j}, \text{ for } j = 1, \ldots, d\}|$ a finite-dimensional parameterization is achieved, thus $R_\phi$ is directly parameterized as complex-valued $(k_{\max} \times v_t \times v_t)$ tensor. Assuming the discretization of domain $D$ into $n \in \mathbb{N}$ points, we find that $v_t \in \mathbb{R}^{n \times d_v}$ and $\mathcal{F}(v_t) \in \mathbb{C}^{n \times d_v}$. Truncating the higher modes yields to $\mathcal{F}(v_t) \in \mathbb{C}^{k_{\max} \times d_v}$ and therefore

$$(R_\phi \cdot (\mathcal{F} v_t))_{k,l} = \sum_{j=1}^{d_v} R_{\phi k,l,j} (\mathcal{F} v_t)_{k,j}, \qquad k = 1, \ldots, k_{\max}, \quad j = 1, \ldots, d_v. \tag{4}$$

When the discretization of the domain $D$ is uniform, $\mathcal{F}$ can be replaced by the Fast Fourier Transform (FFT) method in order to calculate the operation of the kernel integral operator with almost linear complexity (Kovachki et al., 2023).

The methodologies introduced by Li et al. (2022a) and Peng et al. (2022b) involve the utilization of the FNO for predicting 3D turbulent flows. Peng et al. (2023) presented an approach employing the FNO to simulate the 3D dynamic urban microclimate.

**Factorized Fourier Neural Operator (F-FNO).** The Factorized Fourier Neural Operator developed by Tran et al. (2023) incorporates separable spectral layers, refined residual connections, and a combination of different training strategies to enhance performance across a range of various PDEs, surpassing the capabilities of the default FNO. By adding two feedforward layers inspired by the feedforward design used in transformers (Vaswani et al., 2017) and by embedding the residual connection after the non-linearity activation function to preserve more of the layer input, the operator layer in Eq. (2) is changed to

$$v_{t+1}(x) = v_t(x) + \sigma(W_2 \sigma(W_1 \mathcal{K}(a; \phi) v_t(x))), \qquad \forall x \in D. \tag{5}$$

Furthermore, the Fourier transform gets factorized over the problem dimensions $d$, modifying Eq. (3) to

$$(\mathcal{K}(\phi) v_t)(x) = \sum_{i \in d} \mathcal{F}_i^{-1}(R_{\phi i} \cdot (\mathcal{F}_i v_t))(x), \qquad \forall x \in D. \tag{6}$$

The change from $R_\phi$ to $R_{\phi i}$ results in a reduction of the parameter count from $(LH^2 M^d)$ to $\mathcal{O}(LH^2 M d)$, where $H$ denotes the hidden size, $M$ represents the number of top Fourier modes being kept, and $d$ signifies the problem dimension. This optimization is especially useful when addressing higher-dimensional problems, such as simulating the 3D turbulent flow around an object. The incorporation of residual connections after the non-linear activation function preserves more of the layer input and enables the operator to achieve convergence within deep networks.

## 3 METHOD

**Dataset.** The training and testing data are generated using the open source CFD software Open-FOAM. We generate 3D velocity data on a grid of dimensions $108 \times 25 \times 108$ around a cube measuring $12 \times 12 \times 12$ units. The cube maintains a fixed position within the 3D space, remaining unchanged across all samples. The feature edges of the 3D room and the cube are depicted in Fig. A.1. The turbulent flow around the cube is simulated until it reaches a state of convergence. Each sample corresponds to the flow spanning 700 to 800 time steps until convergence is achieved. A total of 118 samples are created, featuring variations in wind speeds and wind directions. These samples encompass wind speeds of 3 m/s, 5 m/s, and 10 m/s. We partition this dataset into 96 samples for training and 18 samples for testing. In all test data, wind directions are deliberately chosen to be distinct from those in the training data, ensuring that the model is solely assessed under entirely unfamiliar scenarios.

**Residual Factorized Fourier Neural Operator (Res-F-FNO).** The architecture of the Res-F-FNO is primary based on the F-FNO framework introduced by Tran et al. (2023). By incorporating residual connections and Fourier factorization, this modified model necessitates fewer parameters than the FNO (Li et al., 2021), which is particularly advantageous in 3D scenarios. Furthermore, those changes also allow F-FNO architectures with significantly more layers to converge. Truncating higher-order Fourier modes result in the loss of small-scale flow structures, leading to a reduction in the precision of the inferred wind field. To address this problem, we add the output of the higher dimensional representation $\mathcal{P}(a(x))$ after the summation of the factorized Fourier transform. Through these residual connections the omitted information from previous layers will be incorporated again, thus improving the overall accuracy of the simulation. We change the Factorized Fourier integral operator defined in Eq. (6) by adding additional residual connections between output of the up projection $\mathcal{P}(x)$ and the output of the summation of the factorized Fourier transforms to:

$$(\mathcal{K}(\phi)v_t)(x) = \mathcal{P}(x) + \sum_{i \in d} \mathcal{F}_i^{-1}(R_{\phi i} \cdot (\mathcal{F}_i v_t))(x), \qquad \forall x \in D. \tag{7}$$

Incorporating these residual connections retains the original number of parameters within the F-FNO framework, consequently preserving the computational performance of the model.

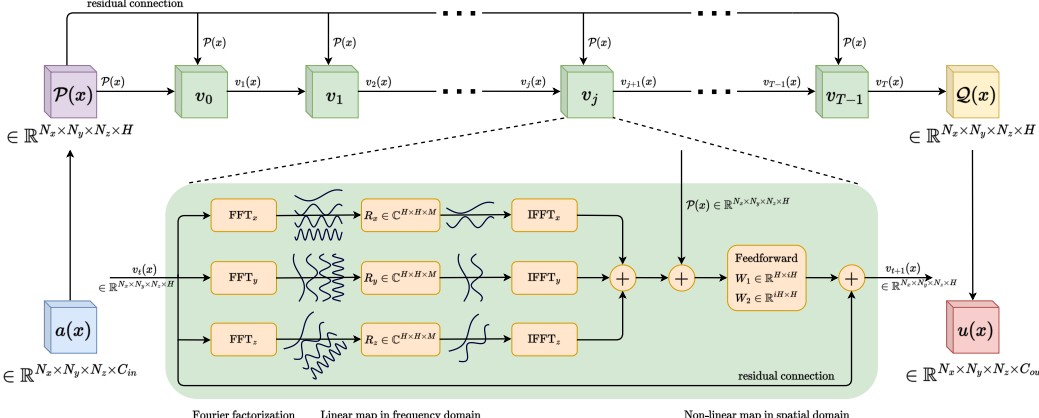

Figure 1: The schematic representation of the Residual Factorized Fourier Neural Operator (Res-F-FNO) architecture utilized for the simulation of 3D turbulent flow around a box. The zoomed-in view of the operator shows the independent processing of each spatial dimension in the Fourier space before their subsequent recombination in the physical space. Subsequently the residual connection is added to the result of the summation (Eq. (7)).

To ensure comparability between the F-FNO and Res-F-FNO architectures, we maintain identical hyperparameters across both models. Both configurations comprise 24 operator layers and accept input tensors with the shape $(N_x, N_y, N_z, C_{in})$, yielding output tensors of dimensions $(N_x, N_y, N_z, C_{out})$. In this context, the dimensions $N_x$, $N_y$, and $N_z$ represent the spatial resolutions within the 3D space, and their values are specified as $N_x = 108$, $N_y = 25$, and $N_z = 108$. $C_{in}$

denotes the input channels, encompassing information for each grid cell, including its classification as an object component (Box=1, Air=0) and the cell center's coordinates $(x, y, z)$. Additionally, wind speeds $v_x$, $v_y$, and $v_z$ at time $t$ are integrated into $C_{\text{in}}$, resulting in an input shape of (108, 25, 108, 7). The output dimensions of the models combine the grid dimension coordinates $(N_x, N_y, N_z)$ and the predicted wind speeds $v_x$, $v_y$, and $v_z$ at time $t+1$, yielding an output shape of (108, 25, 108, 3). The input is projected to the higher dimensional space (108, 25, 108, 64) by a fully connected feedforward network $\mathcal{P}(x)$ before it is used in the first Fourier layer. Accordingly, the output from the final Fourier layer is down-projected from (108, 25, 108, 64) to the output format (108, 25, 108, 3) by a fully connected feedforward network $\mathcal{Q}(x)$. The number of truncated Fourier modes is 32 for the dimensions $N_x$ and $N_z$ and 13 for $N_y$. The described architecture of the Res-F-FNO is also visualized in Fig. 1.

**Skip-corrector.** In the context of simulating PDEs employing data-driven methodologies, such as the FNO framework, wherein successive predictions rely on previous estimates, the prediction error accumulates incrementally for each time step. Considering the simulation of wind fields and their chaotic behaviour over time, this error has the potential to grow exponentially. Reducing the accumulated error for sequential time step prediction remains a challenge for all data-driven methods employed in PDE simulation including wind field predictions (Peng et al., 2022a; 2023; 2022b; Um et al., 2021).

The underlying rationale for the introduction of the skip-corrector is to incorporate an auxiliary solver that employs a coarser temporal discretization scheme. Instead of iteratively solving the governing equations for time instants $t_1, t_2, t_3, \ldots, t_N$, the skip-corrector focuses on the instances $t_1, t_{1+n}, t_{1+2n}, \ldots, t_N$. Let us designate $\hat{v}_t(x)$ as the skip-correctors solution, which is defined only for time instants $t_1, t_{1+n}, t_{1+2n}, \ldots, t_N$. Accordingly, Eq. (5) is reformulated as:

$$v_{t+1}(x) = \begin{cases} \hat{v}_t(x) + \sigma\left(W_2\sigma\left(W_1\mathcal{K}(a;\phi)\hat{v}_t(x)\right)\right) & \text{if } t = 1 + k \cdot n, \\ v_t(x) + \sigma\left(W_2\sigma\left(W_1\mathcal{K}(a;\phi)v_t(x)\right)\right) & \text{else.} \end{cases} \tag{8}$$

A coarser temporal discretization has dual implications. On one hand, the increased interval between discretization points intensifies the complexity of accurately capturing the system's underlying dynamics. Conversely, a less granular temporal resolution mitigates the accumulation of numerical errors propagated by the model. Our empirical observations suggest that, given an optimally selected discretization scheme, the skip-corrector can enhance the predictive accuracy. This is primarily because the reduction in cumulative error tends to outweigh any errors introduced by employing a coarser discretization method. The interaction between the skip-corrector and the subsequent time step prediction models is visualized in Fig. A.2.

The implementation of the skip-corrector leverages the Res-F-FNO architecture, using identical hyperparameters (including the number of layers, Fourier modes, etc.) as those employed by the primary model for predicting subsequent time steps.

**Training strategy.** In the traditional training paradigm for models aimed at simulating PDEs, a dataset is typically generated through numerical solvers, encompassing a wide array of data samples with unique initial conditions. Each sample inherently encompasses of multiple time steps to model the temporal evolution or dynamic behavior of physical phenomena described by the PDE. The dataset is then constructed by rolling out each time step for each sample. In contrast to constructing the dataset using all samples and all time steps, our training methodology involves the selection of a random time step from the time interval of each sample during each iteration. This selected time step serves as the initial condition for predicting the subsequent time step. This approach significantly reduces the duration of a training epoch in comparison to scenarios where the dataset is constructed using all time steps from each sample. This strategy leads to a substantial reduction in the duration of a training epoch when compared to situations where the dataset comprises all time steps from each sample. This is especially beneficial when working with 3D samples.

In addition to our training strategy, we employ the same deep learning techniques as those utilized by Tran et al. (2023). These techniques encompass enforcing the first-order Markov property, implementing the teacher forcing technique, and adding a small amount of Gaussian noise. Notably, one distinction in our approach is the absence of input normalization. We train the models once for 2000 epochs and once for 500 epochs. The learning rate follows a warm-up schedule, gradually increasing over the initial 500 steps until reaching a value of $10^{-4}$, subsequently being reduced using the cosine function. We employ the non-linear ReLU activation function and the Adam optimizer with $\beta_1 = 0.9$, $\beta_2 = 0.99$ and $\epsilon = 10^{-8}$. As evaluation metric and loss function we use the normalized

mean squared error, which is defined as

$$\text{N-MSE} = \frac{1}{B} \sum_{i=1}^{B} \frac{||\hat{\omega} - \omega||_2}{||\omega||_2},$$

where $B$ describes the batch size, $\omega$ the ground truth, $\hat{\omega}$ the prediction and $|| \cdot ||_2$ the l2-norm. The models are implemented in PyTorch and trained on two NVIDIA A100 80GB GPUs.

## 4 EVALUATION

**Comparison against F-FNO.** All the models under consideration, are trained using the randomized time step selection approach for two different durations: 2000 epochs and 500 epochs. The training and testing loss of each model is illustrated in Fig. A.3. The performance of the Res-F-FNO and the F-FNO model in predicting the wind field for the subsequent time step and for the next 100 subsequent time steps is plotted in Fig. 2. Given the utilization of randomly selected time points from the turbulence history as training samples, the plots display the average performance across all test samples when predicting wind fields at various initial time points. This presentation not only facilitates the evaluation of different model performances but also demonstrates the models ability to simulate turbulence from test samples originating at distinct initial time points, despite the randomized initial conditions employed during training.

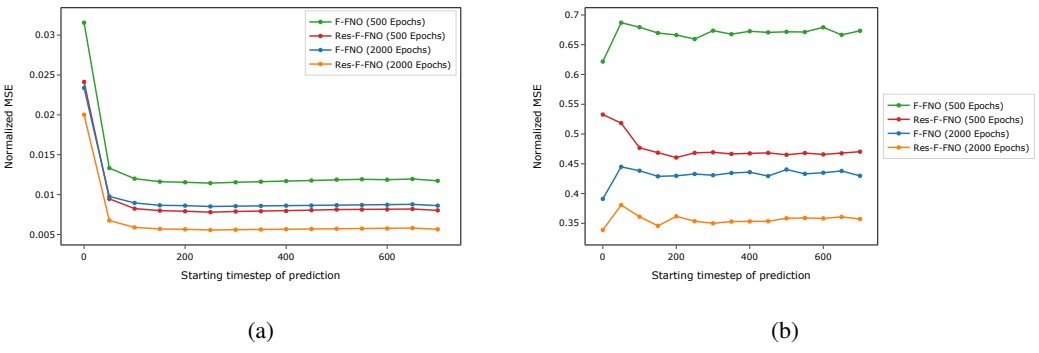

(a)             (b)

Figure 2: Performance comparison of the F-FNO and Res-F-FNO models. In (a), the average N-MSE spanning all test samples for one-time-step predictions across various starting time points is shown. In (b), we show the average N-MSE across all test samples and 100 prediction steps for different initial time points.

After undergoing 2000 training epochs, the F-FNO exhibits an average normalized mean squared error (N-MSE) of 0.0097 when predicting the next time step across all test samples and various initial time points. In contrast, the Res-F-FNO, also trained for 2000 epochs, achieves a notable reduction in error, attaining an average N-MSE of 0.0067 for simulating the subsequent time step. The introduction of residual connections leads to a substantial 30% reduction in error, all while keeping the parameter count unchanged. Furthermore, in the case of models trained over 500 epochs, the Res-F-FNO achieves an N-MSE of 0.0091, which is a remarkable 30% lower than the F-FNO's N-MSE of 0.013. Impressively, the Res-F-FNO trained for 500 epochs even outperforms the F-FNO trained for 2000 epochs by 6%, all while requiring just a quarter of the training time.

In the context of predicting multiple consecutive time steps, the prediction error accumulates with each time step, leading to a notable escalation in the average N-MSE over a span of 100 time steps (Fig. 2b). Specifically, the F-FNO model, trained for 500 epochs, exhibits an average N-MSE of 0.67, while the N-MSE reduces to 0.43 after 2000 epochs of training. In both scenarios, the introduced Res-F-FNO architecture demonstrates its effectiveness in substantially mitigating prediction errors. Following 2000 epochs of training, the Res-F-FNO achieves a 16% reduction in error, decreasing it from 0.43 to 0.36. Additionally, the Res-F-FNO model trained for 500 epochs displays a 28% error reduction, lowering the N-MSE from 0.67 to 0.48 when compared to their respective F-FNO counterparts.

**Effect of additional residual connections.** The truncation of higher-order Fourier modes result in the loss of small-scale flow structures. However, with the incorporation of additional residual connections in the Res-F-FNO architecture, these previously missing structures are reintroduced following the truncation of higher Fourier modes in the Factorized Fourier integral operator. Consequently, this enhancement significantly improves the accuracy of the predicted wind field. Figure 3 presents visualizations of the ground truth, predictions generated by both the F-FNO and Res-F-FNO models, and the absolute error between these predictions and the ground truth for an one-step forecasting. In order to improve the clarity of visualizing turbulent dynamics and the associated absolute error, a clipping operation was applied along the Y-normal plane within the original 3D space, which has dimensions of (108, 25, 108). This clipping procedure effectively reveals the internal structures within the space at a specific height of 6 out of 25 units. This specific height level corresponds to the flow occurring at the mid-height position around the cube, given that the cube object itself has dimensions of (12, 12, 12).

To facilitate a comprehensive evaluation of the models, we calculated and evaluate the absolute error between their predictions and the ground truth. Particularly, we focused on assessing the models' capability to resolve small deviations falling within the range of 0.0 to 0.1 m/s. The integration of additional residual connections in the Res-F-FNO architecture yields a notable reduction in these minor deviations compared to the outcomes of the F-FNO model. Consequently, the Res-F-FNO exhibits an enhanced ability to predict turbulence with greater accuracy, as evidenced by the reduction in N-MSE.

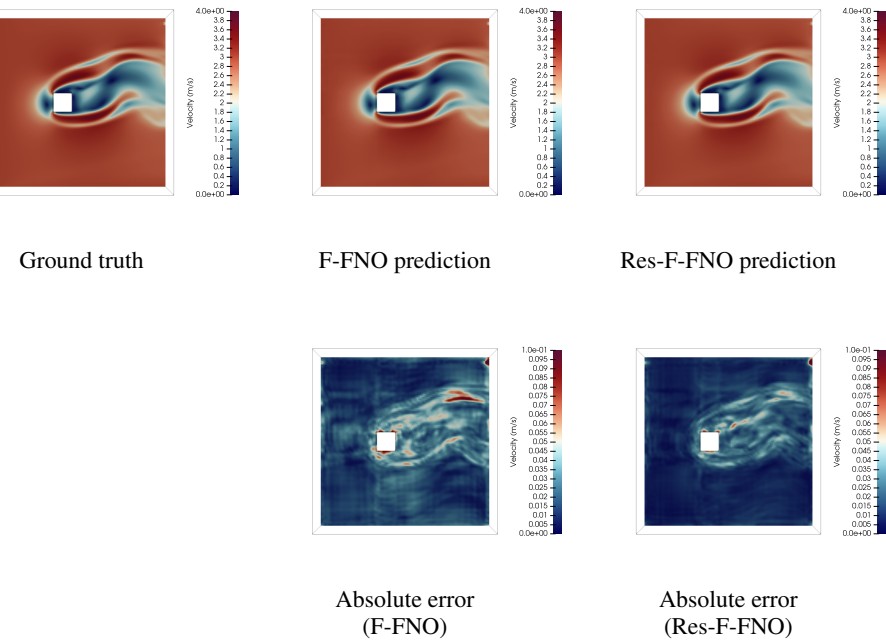

Figure 3: One-step prediction comparison: We contrast the 3D flow prediction for a single time step between the F-FNO and Res-F-FNO models. The visualization in the second row illustrates the absolute error between the ground truth and the prediction.

**Effect of the skip-corrector.** In order to forecast multiple consecutive time steps, both the Res-F-FNO and F-FNO models take the previous prediction as ground truth for estimating turbulence in the subsequent time step. This approach results in an accumulation of prediction errors over a sequence of time steps, leading to a substantial increase in the N-MSE throughout the prediction of multiple time steps.

As depicted in Fig. 4a, all models exhibit a consistently low N-MSE during the initial time step prediction. Specifically, the F-FNO model, trained over 500 epochs, yields an average N-MSE of 0.013 across all test samples for this initial prediction, while the F-FNO model trained over 2000 epochs achieves an average N-MSE of 0.0095. In contrast, the corresponding Res-F-FNO models

demonstrate an even better performance, with N-MSE values of 0.0091 and 0.0065, respectively, representing a notable enhancement of 30% for each model.

When predicting multiple consecutive time steps, prediction errors cumulatively impact each step. For instance, at the 50th time step prediction, the F-FNO model trained over 500 epochs records an average N-MSE of 0.54, whereas the model trained over 2000 epochs exhibits a reduced N-MSE of 0.35. In contrast, the Res-F-FNO models exhibit further improvement with average N-MSE values of 0.43 and 0.32, corresponding to 20% and 8% enhancements over their respective F-FNO counterparts.

Extending the analysis to the 100th time step prediction, the N-MSE rises to 1.61 for the F-FNO model trained over 500 epochs and 1.06 for the model trained over 2000 epochs. In contrast, the Res-F-FNO models exhibit superior predictive capabilities, achieving a 30% and 11% reduction in error, resulting in corresponding N-MSE values of 1.12 and 0.94, respectively.

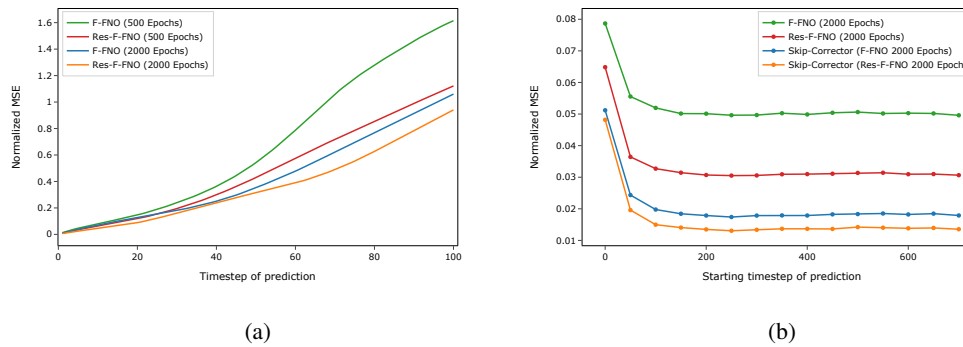

(a)            (b)

Figure 4: Performance comparison of the F-FNO, Res-F-FNO, and skip-corrector models. In (a), the average N-MSE for each of the 100 prediction steps is displayed across all test samples and starting points. In (b), we present the average N-MSE across all test samples, considering various initial time points for predicting the 6th consecutive time step.

The primary objective of the skip-corrector is to substantially mitigate the cumulative error that occurs during the prediction of numerous consecutive time steps. Specifically, it is trained simulate the wind field $u(x)_{t+n}$, at a considerably extended time interval based on the input variable $a(x)_t$. In this context, $t$ represents the initial temporal point, while $n$ denotes the number of intermediate time steps spanning between the input and the desired output. Our study involves the utilization of two distinct architectural implementations for the skip-corrector: firstly, employing the F-FNO architecture, and secondly, employing the Res-F-FNO architecture. A comparative analysis of these two approaches is presented. Both models are trained for 2000 epochs, with the specific objective of predicting the state $u(x)_{t+6}$, relying on the input $a(x)_t$. The training and testing loss for each model is visualized in Fig. A.4.

In Fig. 4b, the average N-MSE for predicting the 6th time step across all samples is illustrated. Initial conditions for those predictions were established using wind fields at time points $(0, 50, 100, 150, ..., 700)$. Both the skip-corrector based on the F-FNO architecture and the skip-corrector based on the Res-F-FNO architecture exhibit superior accuracy in predicting the 6th time step when compared to the F-FNO and Res-F-FNO models. Notably, the cumulative error incurred during the prediction of 6 consecutive time steps is markedly higher than the N-MSE recorded when directly predicting the 6th time step using the skip-corrector approach.

Specifically, the F-FNO model yields an N-MSE of 0.052 for the prediction of the 6th consecutive time step, while the Res-F-FNO model achieves an N-MSE of 0.034. This represents a significant 34% reduction in error when utilizing the Res-F-FNO architecture. The skip-corrector employing the F-FNO architecture attains an average N-MSE of 0.021 for direct prediction of the 6th time step, reducing the error by a substantial 59% in comparison to the F-FNO model and 38% in comparison to the Res-F-FNO model. Furthermore, the skip-corrector utilizing the Res-F-FNO architecture achieves an average error of 0.016 in the direct prediction of the 6th time step. This results in a substantial 69% reduction in N-MSE compared to the F-FNO model, a 52% reduction compared to

the Res-F-FNO model, and a 23% reduction in comparison to the N-MSE associated with the skip skip-corrector implemented by the F-FNO architecture.

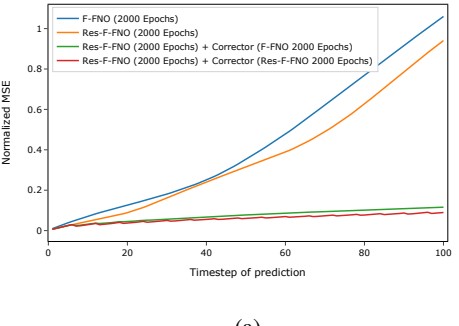
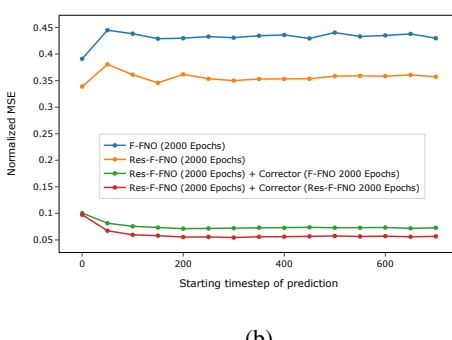

(a)                  (b)

Figure 5: Impact of the skip-corrector when predicting 100 consecutive time steps. In (a), the average N-MSE for each of the 100 prediction steps is exhibited across all test samples and starting points. In (b), we illustrate the average N-MSE across all test samples, encompassing different initial time points for predicting 100 consecutive time steps.

The integration of the skip corrector yields a substantial reduction in cumulative error when predicting 100 consecutive time steps (Fig. 5a). Notably, while the N-MSE for the prediction of the 50th time step stands at 0.35 and 0.32 for the F-FNO and Res-F-FNO models, respectively, these values can be markedly decreased to 0.078 and 0.06 through the combination of Res-F-FNO with the skip corrector implemented by the F-FNO architecture, and 0.06 when using the skip corrector represented by the Res-F-FNO architecture. This corresponds to a remarkable enhancement of 77% and 82% compared to the F-FNO model and 75% and 81% compared to the Res-F-FNO model. In forecasting the 100th time step, the F-FNO model exhibits an average N-MSE of 1.06, while the Res-F-FNO architecture achieves a lower N-MSE of 0.94. The incorporation of the skip corrector, implemented by the F-FNO architecture, results in a substantial error reduction to 0.12. This represents an enhancement of 88% and 87%, respectively.

When considering the average N-MSE across all samples and various initial conditions at different time points $(0, 50, 100, 150, ..., 700)$, the F-FNO model yields an N-MSE of 0.43, while the Res-F-FNO model achieves an N-MSE of 0.35. Integration of the skip-corrector, implemented by the F-FNO architecture, leads to a notable reduction in the average error, bringing it down to 0.075. This represents a substantial improvement of 82% and 78%, respectively. Furthermore, the utilization of the skip corrector embedded by the Res-F-FNO architecture results in a further reduction of the error to 0.06. This corresponds to a significant enhancement of 86% and 82%, respectively. Additionally, when employing the skip corrector which utilizes the Res-F-FNO architecture, the error is further reduced to 0.09, corresponding to an even more substantial reduction of 91% and 88%, respectively (Fig. 5b).

## 5 CONCLUSION

In this study, we have demonstrated that the Res-F-FNO model, coupled with the skip-corrector concept, exhibits the capability to forecast turbulent flow patterns around a cube over a span of 100 time steps with an average N-MSE of less than 7%.

For future work, we are interested to explore how effective this approach can generalize to various objects and shapes, provided that the dataset is expanded accordingly. Furthermore, it would be interesting to investigate to what extent the cumulative error can be further reduced by incorporating attention approaches or physical constraints.

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

# A APPENDIX

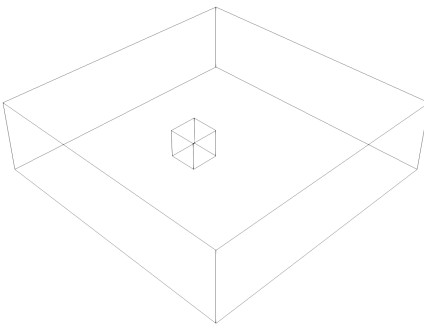

Figure A.1: Illustration depicting the feature edges within the 3D space and the cube it encompasses. The room's dimensions are $108 \times 25 \times 108$, while the cube's dimensions are $12 \times 12 \times 12$.

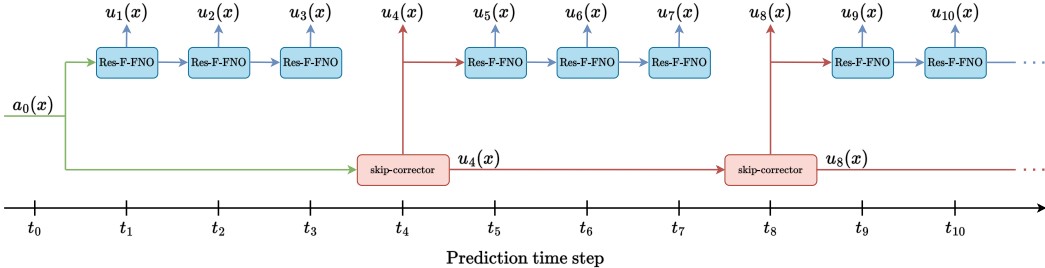

Figure A.2: Visual representation of the interaction between the Res-F-FNO and the skip-corrector in predicting 10 consecutive time steps. In this particular scenario, the skip-corrector is trained to specifically predict the 4th successive time step based on the current input. The process commences with the input variable $a(x)$ serving as the initial condition at time $t_0$, from which the subsequent 10 time steps are simulated.

The prediction is executed by the Res-F-FNO model for the first 3 time steps, where each prediction employs the preceding prediction as input to determine the subsequent state. The prediction of the 4th time step, is performed by the skip-corrector. This skip-corrector receives, as input, the state from 3 time steps prior, in this instance the wind field at time $t_0$, and subsequently predicts the state at $t_4$. This prediction then serves as input for the Res-F-FNO model, which proceeds to determine the states at $t_5$, $t_6$, and $t_7$ based on the preceding time step.

Given the specific training of the skip-corrector, which is configured to predict every 4th time step, it forecasts the state at time $t_8$. The skip-corrector uses the wind field data from time $t_4$ as its input. The states at time $t_9$ and $t_{10}$ are subsequently determined by the Res-F-FNO model, leveraging the preceding states at $t_8$ and $t_9$ as their input respectively.

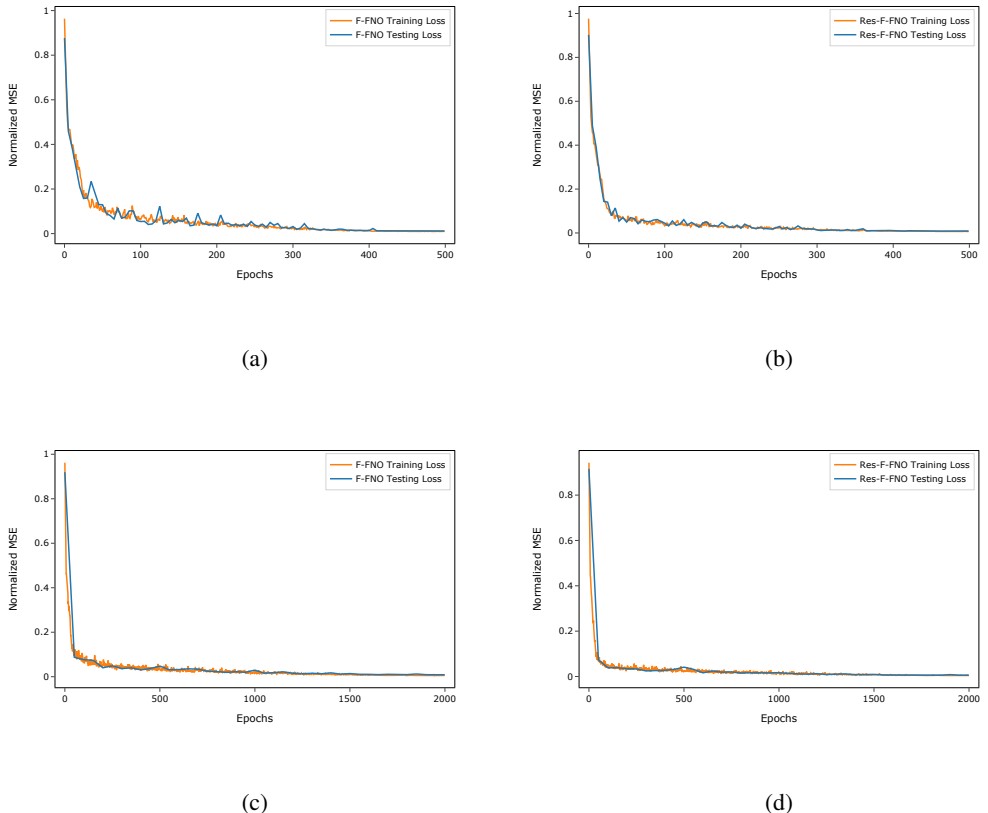

(a)

(b)

(c)

(d)

Figure A.3: Training and testing curves for the F-FNO and Res-F-FNO models. In (a), we present the training and testing curve for the F-FNO model over 500 epochs. In (b), the training and testing curve for the Res-F-FNO model, trained for 500 epochs, is displayed. In (c), we depict the training and testing curve for the F-FNO model, trained for 2000 epochs. In (d), the training and testing curve for the Res-F-FNO model, trained for 2000 epochs, is visualized. All training curves exhibit stability, and each model successfully converged.

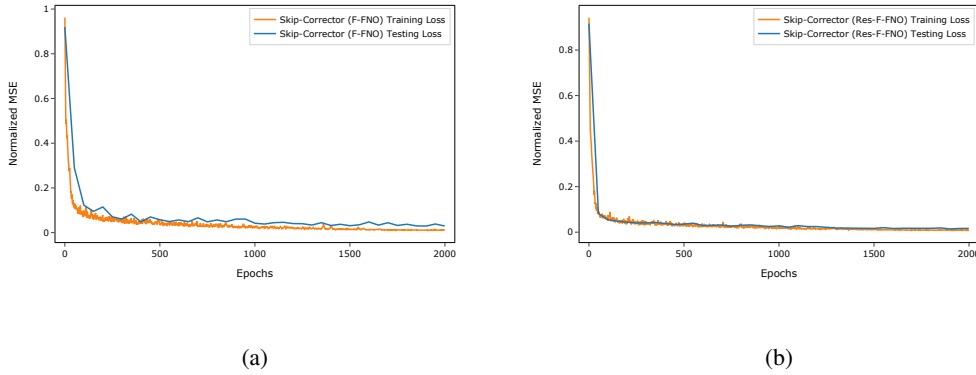

(a)

(b)

Figure A.4: Training and testing curves for the skip-corrector models. In (a), we present the training and testing curve for the skip-corrector which utilizes the F-FNO architecture. In (b), the training and testing curves for the skip-corrector which is implemented by the Res-F-FNO architecture is visualized.

