# OpenReview forum: "Residual Factorized Fourier Neural Operator for simulation of three-dimensional turbulence"
_ICLR.cc/2024/Conference — Submitted to ICLR 2024_

### Official Review · Reviewer_zqsB · 2023-10-13

**Soundness:** 2 fair
**Presentation:** 2 fair
**Contribution:** 2 fair
**Rating:** 3
**Confidence:** 3

**Summary:**

The authors propose a FNO based architecture which makes use of additional residual connections and the factorized FNO introduced in [1].  Moreover, a skip-corrector is introduced which essentially takes larger timesteps, which the authors argue is beneficial as it avoids accumulated errors. This architecture is trained on 3D turbulent flow around a cube, with ablations showing the improvements of each of these measures.

**Strengths:**

* this dataset is a custom dataset and not many ML-based approaches have attempted to solve three-dimensional fluid flow problems.
* the discussion on accumulated errors vs. taking large timesteps is intriguing.

**Weaknesses:**

* the architecture is hardly novel - apart from being an adapted version of [1], in my view this arrangement of residual connections (around the convolution + around the FNO layer + one large residual connection) have also been used in [2] and [3].
* If I have misunderstood the novelty of the residual connections/missed sth., I would encourage the authors to clarify this better in their paper.
* the design of the skip-corrector reminds me of the hierarchical time stepping scheme utilised in [4]. While the authors of [4] make similar arguments about accumulated errors vs. the error of making larger timesteps, this was not the main focus of their paper. I would have welcomed a more detailed analysis on the benefit of the skip-corrector here. Especially on the influence of the parameter $n$ - the timestep of the skip-corrector
* Overall, the experimental evaluation leaves a lot to be desired - experiments were performed on a single dataset with a single baseline (F-FNO). It is unclear to me how hyperparameters play into this. How does a regular FNO or a tensorized one [3] perform on this dataset?

#### References
[1] Alasdair Tran, Alexander Mathews, Lexing Xie, and Cheng Soon Ong. Factorized fourier neural operators, 2023.

[2] Bonev B., Kurth T., Hundt C., Pathak, J., Baust M., Kashinath K., Anandkumar A.; Spherical Fourier Neural Operators: Learning Stable Dynamics on the Sphere; arXiv 2306.0383, 2023.

[3] Neuraloperator library: https://github.com/neuraloperator/neuraloperator

[4] K Bi, L Xie, H Zhang, X Chen, X Gu, Q Tian; Pangu-weather: A 3d high-resolution model for fast and accurate global weather forecast; arXiv preprint arXiv:2211.02556, 2022.

**Questions:**

* You argue that it is beneficial to take large timesteps thus avoiding accumulation errors. Why not take as large of a timestep possible in this case?
* Surely there must be a "golden middle"? A study on the optimal skip timestep to take would be quite informative.

---

### Official Review · Reviewer_UKe3 · 2023-10-27

**Soundness:** 2 fair
**Presentation:** 1 poor
**Contribution:** 1 poor
**Rating:** 3
**Confidence:** 5

**Summary:**

This submission added an MLP in the skip-connection in the FNO spectral conv layer based on the Factorized FNO layer proposed in Tran et al. ICLR 2023, and used a "corrector" (with the same architecture) to restart the temporal prediction every few time steps. The method is then tested on a classical CFD benchmark problem. Due to a limited contribution from either theoretical or experimental aspect and lacking a detailed data description, this submission currently does not meet the bar of ICLR.

**Strengths:**

- This submission investigated an important topic in the cross section of CFD and data-driven methods. The simulation of NSE with high Reynolds number, especially in 3D, still remains to be a great challenge in the numerical PDE community. The benchmark problem featured (flow around a square cylinder confined in a channel) is also an important problem in CFD in terms of theoretical, experimental, and also for numerical simulation aspects. In fact, the obstacle being cubic makes this problem even harder (to simulate the vortices' interaction in both lateral and vertical direction).
- The phase variable (obstacle or air) as an input feature channel is a good practice (and references should be given such as [KLY] below).
- A training strategy similar to DDPM is used temporally (however, no references are given on page 5).

[KLY]: Y. Khoo, J. Lu, L. Ying, arXiv:1707.03351

**Weaknesses:**

- The author claimed the modification of FNO by adding two FFN layers are inspired by the Transformers. However, it has long been acknowledged that FNO resembles Transformers, and the skip-connection can be added, for example in [GMZ] (but the authors there did not make it a big deal and just updated the code, please check the official FNO GitHub repo `master` branch commit `9bc9516`).
- Note the applying the tensorized architectures in Transformers to tackle the 3D NSE in the turbulent regime has been explored in [LSF], and Transformers for operator learning have a long list of references in NeurIPS/ICLR/ICML, but none of these are mentioned.
- This submission copies the FNO paper's presentation on the neural architecture on the first half of the page 3, and therein the letter $t$ corresponds to the layers. Yet later, on page 5, $t$ now denotes the time.
- Similarly, notational discrepancy applies to the $x$ variable. Throughout the presentation of the neural architecture, $x\in D$, meanwhile $x$ also denotes the $x$-direction.
- Apparently, the FFN's weight matrices $W_1$'s and $W_2$'s dimensions in Figure 1 are wrong.
- In the channeled obstacle flow problem, the outflow boundary (the right side of the region in Figure 3) needs a special boundary condition. This goes unmentioned at all. In FNO and all its variants with no modification in this regard, periodical BC is used (and the vortices will be bounced back from the right boundary, not like what Figure 3 shows). This paper gives no specific change in this regard.
- Continue the point above, in this channeled obstacle flow problem, the real difficulty is whether a numerical scheme can handle the von Kármán vortices elegantly. Therefore, the computational domain reserves enough room in the outflow direction with respect to the obstacle. Yet, this paper just uses a square domain when viewed from the cross-section perspective. For example, please check classical papers on this issue such as Sohankar et al, Int. J. Numer. Methods Fluids 1998.
- The study of the effect of additional residual connections on page 7 is way too superficial by just comparing the output to meet ICLR caliber, please check how ablation study for adding skip-connection should be conducted, for example in [HZRS] or more theoretically such as [OP] and [HWTZ].
- The error convergence just shows that after maybe 100 epochs, the 20 times more computational cost to train the model gives marginal improvement in accuracy. How to justify this extra cost? If such computational resources can be exploited why not just use DNS?

----

[GMZ]: Efficient token mixing for Transformers via adaptive Fourier neural operators, ICLR 2022.

[LSF]: Scalable Transformer for PDE Surrogate Modeling, arXiv:2305.17560.

[HZRS]: He et al., arXiv:1603.05027.

[OP]: Orhan and Pitkow, ICLR 2018.

[HWTZ]: Huang et al., Why Do Deep Residual Networks Generalize Better than Deep Feedforward Networks? NeurIPS 2021.

**Questions:**

- The presentation on page 4 to 5 is unclear of how the skip-connection is exploited and what is skip-connected. For example, on page 4 it says "add the output $\mathcal{P}(a(x))$" yet in Figure 1 it is "$\mathcal{P}(x)$". If it is the case of the latter, that $\mathcal{P}(\cdot)$ is an MLP that maps $x\in \mathbb{R}^3$ to $\mathbb{R}^3$ then this is just a function learner not an operator learner. Otherwise the presentation is plainly wrong. On page 5, it says "the input is projected to a higher dimensional space by an FFN $\mathcal{P}(x)$", is $\mathcal{P}(x)$ an FFN? What is its domain and range?
- The data are said to be generated by OpenFOAM. However, no information is given for the equation used. Is it RANS or DNS? Reynolds number is not given either. The reason to present this is that the NSE benchmark in the FNO paper (and subsequent work) uses the streamfunction-vorticity formulation that is not attainable in 3D.  It says "until it reaches a state of convergence", but convergence in what? Is the flow becoming a steady-state? How the wind directions (I am assuming the authors meant to say the inflow boundary) are chosen?
- The presentation of the skip-corrector is unclear. How to optimally select a discretization scheme?

---

### Official Review · Reviewer_8Jew · 2023-10-30

**Soundness:** 2 fair
**Presentation:** 2 fair
**Contribution:** 1 poor
**Rating:** 3
**Confidence:** 4

**Summary:**

The authors introduce a neural operator that learns to simulate 3d turbulence flows. The main adaptations includes additional residual connections that preserve reference to small-scaled features amidst FNO's mode truncation, and a skip-corrector that brings useful information from an auxiliary solver such that the neural operator does not have to learn it. The authors perform evaluation on a 3d flow around a box and present ablation studies showing the effects of the introduced adaptations.

**Strengths:**

* The authors study a 3-dimensional flow problem, which has good complexity and significance, although some important details from data generation is not included.

**Weaknesses:**

* The novelty in the idea, i.e. introducing additional residual connections and incorporating outputs from coarse solvers, is relatively marginal to me. Both of these have been widely used in various contexts and the latter seems to even create unphysical features (c.f. questions below) in the rollout.
* Clarity - some important details missing including the setup of the turbulent flow problems and information on the auxiliary solver employed for the skip corrector (see questions below).
* Baselines - the proposed method is only compared against a factorized FNO model. The authors also include versions trained with 500 and 2000 epochs but it is unclear to me why the former is worth comparing as it is not fully trained.
* Evaluation is lacking - only normalized mean squared errors are shown. At the very least one would expect spectral metrics and example rollouts to be shown.

**Questions:**

* I do not quite follow "selection of a random time step from the time interval of each sample during each iteration" mentioned in the training strategy section (page 5, second last paragraph). Is this equivalent to just picking a random subsection of a long trajectory to use for training (which is quite standard and I'm unsure why it's worth highlighting)?
* Does the way that the skip-corrector is set up not result in discontinuities on the steps it kicks in? This is even somewhat visible in Figure 5(a), where the red error curve has kinks every few steps.
* What does "the turbulent flow ... is simulated until it reaches a state of convergence" (page 4, first paragraph) mean? Do you mean statistical equilibrium? Also what are the initial conditions, boundary conditions, forcing and Reynolds number of the flow respectively?
* In Figure 2a, 4b and 5b, why does the model perform better when the starting time step of prediction is small?
* Comparing Figure 4a and 5a - it seems that the corrector makes a drastic difference in improving the metrics. This naturally prompts the question of whether the auxiliary solver is already providing most of the prediction skills and how much Res-F-FNO contributes? It would be useful to include the setup, error levels and stability characteristics of the auxiliary solver for reference.
* How is the long term stability of the model when rolled for an indefinite number of time steps?

---

### Official Review · Reviewer_P4Hh · 2023-10-31

**Soundness:** 2 fair
**Presentation:** 2 fair
**Contribution:** 1 poor
**Rating:** 3
**Confidence:** 4

**Summary:**

This paper explores the introduction of a residual connection to the Factorized Fourier Neural Operator and evaluates this idea on a custom dataset of 118 simulations of turbulent flow around a 3D cube. It further explores the effect of simulating at a coarser time resolution ("skip corrector") and randomly subsampling the dataset each epoch to decrease the time it takes to train for one epoch ("innovative training methodology").

**Strengths:**

The paper introducers a small novel dataset and compares against a baseline on a hold-out test set.

**Weaknesses:**

- The skip corrector contribution is misrepresented. Rather than correcting the output of the fine-grained F-FNO, it ignores the fine-grained predictions and simply solves the simulation in isolation at a coarser time resolution. The fine-grained F-FNO now just serves as a method to increase the temporal resolution of the coarse grained model. This is not correcting the fine-grained model, but instead augmenting the coarse grained model. Any error introduced by the coarse-grained model is propagated by the fine-grained model. This idea is orthogonal to the introduced residual connections (often called "skip connections" to further muddy the distinction) and distracts from the core contribution. If simulation at a coarse-grained time resolution is suitable for a problem, what is the point of additionally upsampling the temporal resolution?
- The evaluation is limited by choosing only one system under limited starting condition variations. The PDE solving FNO literature has a wide variety of benchmarks, and the experimental section should be extended to provide evidence that the Res-F-FNO architecture is superior to the F-FNO model across more than one problem. It's not clear how the Res-F-FNO performs compared to F-FNO on a real-world generalization task.
- The merit of the proposed novel training strategy is not explained. The definition of what constitutes an epoch is arbitrary and it's unclear if the new strategy reduces the number of FLOPs required to train the model to convergence, or by proxy, the number of samples the model sees.

**Questions:**

- The choice of evaluating models at 500 and 2000 training epochs seems arbitrary. It would be better to choose a convergence metric and compare both models at their optimum. If not, can you justify the choice of these steps, and does this choice benefit one model over the other?
- Have you studied the discontinuities between the coarse-grained predictions and the adjacent last fine-grained predictions?

---

### Meta-Review · Area_Chair_RnQP · 2023-12-05

**Metareview:**

The authors propose a Factorized Neural Operator (FNO)-based architecture that extends the original FNO by incorporating an additional residual connection coupled with a small MLP corrector. They argue that this new architecture enables them to use larger timesteps, which in turn prevents error accumulation. The architecture is benchmarked on 3D turbulent flow around a cube. The authors perform ablation studies to demonstrate the improvements achieved by each component.

The reviewers agree that the method represents a minor modification of an existing approach. They find the evaluation, including the choice of baseline, to be overly narrow. Additionally, they deem the ablation studies inadequate and the computational savings unclear. The authors did not offer any feedback during the rebuttal stage, leaving all concerns unaddressed.

Due to the lack of novelty, the narrow evaluation, and the unconvincing results, I recommend to reject.

**Justification For Why Not Higher Score:**

The novelty is marginal, the architecture was tested in a single example, and only against other factorized FNO method.

**Justification For Why Not Lower Score:**

N/A

---

### Decision · Program_Chairs · 2024-01-16

Reject